# Spatio-temporal clustering of Mountain-type Zoonotic Visceral Leishmaniasis in China between 2015 and 2019

Yuwan Hao[1], Xiaokang Hu[1], Yanfeng Gong[1], Jingbo Xue[1], Zhengbin Zhou[1], Yuanyuan Li[1], Qiang Wang[1], Yi Zhang[1,2], Shizhu Li[1,2]*

1 National Institute of Parasitic Diseases, Chinese Center for Disease Control and Prevention; National Center for Tropical Diseases Research; WHO Collaborating Centre for Tropical Diseases; National Center for International Research on Tropical Diseases, Ministry of Science and Technology; Key Laboratory of Parasite and Vector Biology, Ministry of Health, Shanghai, China, 2 School of Global Health, Chinese Center for Tropical Diseases Research-School of Medicine, Shanghai Jiao Tong University, Shanghai, China

* lisz@chinacdc.cn

**Data Availability Statement:** Not all data can be shared publicly because the data includes individual information, such as the name, gender, age, address, ID number, and so on, and the

## Abstract

With several decades of concerted control efforts, visceral leishmaniasis(VL) eradication had almost been achieved in China. However, VL cases continue to be detected in parts of western China recent years. Using data of reported cases, this study aimed to investigate the epidemiology and spatio-temporal distribution, of mountain-type zoonotic visceral leishmaniasis (MT-ZVL) in China between the years 2015 and 2019. Epidemiological data pertaining to patients with visceral leishmaniasis (VL) were collected in Gansu, Shaanxi, Sichuan, Shanxi, Henan and Hebei provinces between the years 2015 and 2019. Joinpoint regression analysis was performed to determine changes in the epidemic trend of MT-ZVL within the time period during which data was collected. Spatial autocorrelation of infection was examined using the Global Moran's $I$ statistic wand hotspot analysis was carried out using the Getis-Ord $Gi^*$ statistic. Spatio-temporal clustering analysis was conducted using the retrospective space-time permutation flexible spatial scanning statistics. A total of 529 cases of MT-ZVL were detected in the six provinces from which data were collected during the study time period, predominantly in Gansu (55.0%), Shanxi (21.7%), Shaanxi (12.5%) and Sichuan (8.9%) provinces. A decline in VL incidence in China was observed during the study period, whereas an increase in MT-ZVL incidence was observed in the six provinces from which data was obtained ($t = 4.87$, $P < 0.05$), with highest incidence in Shanxi province ($t = 16.91$, $P < 0.05$). Significant differences in the Moran's $I$ statistic were observed during study time period ($P < 0.05$), indicating spatial autocorrelation in the spatial distribution of MT-ZVL. Hotspot and spatial autocorrelation analysis revealed clustering of infection cases in the Shaanxi-Shanxi border areas and in east of Shanxi province, where transmission increased rapidly over the study duration, as well as in well know high transmission areas in the south of Gansu province and the north of the Sichuan province. It indicates resurgence of MT-ZVL transmission over the latter three years of the study. Spatial clustering of infection was observed in localized areas, as well as sporadic outbreaks of infection.

information is strictly protected and not permitted to be distributed without approval from the government. Therefore, the case information cannot be shared after analysis. However, if any researchers are interested in accessing the data without commercial purpose, they can contact Dr. Shang Xia, who will reach out to the government for permission. The contact information: Dr Shang Xia, sxia@nipd.chinacdc.cn 008621-54241570 Chief of Informatics Center Department National Institute of Parasitic Diseases, Chinese Center for Disease Control and Prevention No. 207, Ruijin Er Road, Shanghai 200025, China.

**Funding:** This study was supported by the National Special Science and Technology Project for Major Infection Diseases of China (No. 2016ZX10004222-004) (YH, XH, YG, QW, SL), the scientific investigation on regional climate-sensitive diseases in China (No.2017FY101203) (JX, ZZ, YL, YZ), the Fifth Round of Three-Year Public Health Action Plan of Shanghai (No. GWV-10.1-XK13) (YH, JX, SL). The funders had no role in study design, data collection and analysis, decision to publish, or preparation of the manuscript.

**Competing interests:** The authors have declared that no competing interests exist.

## Author summary

*Leishmania* parasite. It was defined as a neglected tropical disease (NTD) by the World Health Organization (WHO) since 2010. The eradication of VL had almost been achieved in the country since 1960s'last century but the parts of western China. Although the numbers of annual reported cases of VL declined, the mountain-type zoonotic visceral leishmaniasis (MT-ZVL) continued to increase in recent years. In this study, the epidemiological characters and spatio-temporal distribution of MT-ZVL were investigated in China between the years 2015 and 2019. A total of 529 cases of MT-ZVL were reported in six provinces and predominantly in Gansu (55.0%), Shanxi (21.7%), Shaanxi (12.5%) and Sichuan (8.9%) provinces. Significant differences in the Moran's I statistic were observed, indicating spatial autocorrelation in the spatial distribution of MT-ZVL. Spatio-temporal hotspot analysis revealed clustering of infection cases in the Shaanxi-Shanxi border areas and eastern of Shanxi province, as well as in the south of Gansu province and the north of the Sichuan province. Therefore, the reinforcement of VL control in conventionally high-risk areas, attention to areas where VL re-emergence is likely, timely survey of vectors, assessment of transmission risk, and targeted interventions are strongly recommended to reduce risk of MT-ZVL infection.

## Introduction

Visceral leishmaniasis (VL), also known as kala-azar, is a zoonotic infectious disease caused by the protozoan *Leishmania* parasite and transmitted by the bite of infected sandflies [1]. Currently, this zoonosis is prevalent in 88 countries across East Africa, South Asia, South America and the Mediterranean [2]. Global incidence of infection is estimated at 200 to 400 thousand cases each year, with approximately 60 thousand VL attributed deaths due to failure in timely treatment [3]. Consequently, VL related mortality ranks second only to malaria among all parasitic diseases in terms of the mortality [4]. In 2010, VL was defined as a neglected tropical disease (NTD) by the World Health Organization (WHO). NTDs are a group of parasitic and bacterial diseases intimately linked to poverty and affecting more than one billion people worldwide annually [5–6]. Although VL transmission in China was once prevalent across 16 provinces north of the Yangtze River, eradication had almost been achieved at the beginning of the 1960s following several decades of concerted control efforts [7]. Currently however, VL cases continue to be detected in parts of western China, including Kashgar in Xinjiang, the southern Gansu province and northern Sichuan province, with localised clustering of VL occasionally reported [8].

The three main classifications of VL infection in China are anthroponotic visceral leishmaniasis (AVL), mountain-type zoonotic visceral leishmaniasis (MT-ZVL) and desert-type zoonotic visceral leishmaniasis (DT-ZVL) [9]. Among these, significant variations exist in the in geographical predominance and ecology of transmission, at-risk populations and vector species [10]. AVL and DT-ZVL transmission occurs predominantly in Xinjiang, while MT-ZVL is prevalent across other areas of China, including parts of Gansu, Sichuan, Shaanxi, Shanxi, Henan and Hebei provicce, which locating in the extension region of Loess Plateau. Since 2016, VL has been included in the National Control Program for Major Parasitic Diseases in China (2016–2020) and given a high priority of management. As a result, transmission control of VL has been achieved in the country, with the number of cases declining over the same time period [11]. Despite this, cases of MT-ZVL have

increased each year where MT-ZVL is endemic, and resurgence and clustering of MT-ZVL has been reported in multiple endemic foci of China. In 2019, with a total of 52 MT-ZVL cases detected in Shanxi province, an incidence much higher than the mean provincial prevalence reported between 2014 through 2018, and a 13-fold increase compared with cases detected in 2014 (4 cases). Also of importance was the observed transmission of MT-ZVL to neighboring regions [11–12].

Experiences and lessons learned from the Chinese VL control program have demonstrated that the VL transmission is likely to rebound once the control efforts are weakened [13]. As transmission of VL and the distribution of sandfly populations are greatly affected by natural, biological and social factors [14–15], timely identification of VL cases, and consolidated development and optimization of control strategies, are needed for the interruption and elimination of VL transmission. [16–17]. This study aimed to investigate the epidemiology, and assess the temporal and spatial distribution pattern, of MT-ZVL in China between 2015 to 2019, in order to provide insights into the development of targeted interventions for MT-ZVL.

## Materials and methods

### Data acquisition

Visceral leishmaniasis case data, reported in Gansu, Shaanxi, Sichuan, Shanxi, Henan and Hebei provinces between 2015 and 2019 were obtained from the National Notifiable Communicable Disease Reporting System [18]. Counties from each of the six provinces where MT-ZVL cases were detected were selected as sampling sites, and longitude and latitude coordinates determined for each site.

### Analysis of changes in epidemic trend of MT-ZVL

All epidemic data pertaining to MT-ZVL were loaded into Microsoft Excel 2013. The epidemic trend of MT-ZVL were analyzed using descriptive epidemiology approach and the long-term changing of MT-ZVL incidence were tested using Joinpoint Model of the Joinpoint Regression Program (Version4.3.1). The *T*-tests were used to determine whether there is significant difference in the long-term changing of the incidence within a certain period of time [19], the long-term trend in linear segments were described according to the best fitting results, and Annual Percentage Change (APC) values were calculated [20].

### Spatial autocorrelation analysis

Spatial autocorrelation is defined as the correlation of values of a single variable at different geographical locations using a measurement of spatial clustering based on feature locations and attribute values [21]. Spatial autocorrelation and hotspot analysis were performed using the global Moran's *I* and Getis-Ord $G_i^*$ statistics, respectively, in ArcGIS software, version 10.3 [22]. The Global Moran's *I* statistic estimates the overall degree of spatial correlation for a dataset [6–7], and is calculated using the following formula:

$$I = \frac{n \sum_{i=1}^{n} \sum_{j=1}^{n} w_{ij}(x_i - \bar{x})(x_j - \bar{x})}{\sum_{i=1}^{n} \sum_{j=1}^{n} w_{ij} \sum_{j=1}^{n} (x_i - \bar{x})^2}$$

where *I* is indicative of the Moran's *I* statistic, with values ranging from -1 (perfect dispersion) to 1 (perfect correlation). Negative values indicate negative spatial autocorrelation, positive values indicate positive spatial autocorrelation, and a value of zero value indicates a random spatial pattern (no spatial correlation).

The Getis-Ord $G_i^*$ statistic, a spatial autocorrelation index based on a weighted distance matrix, ascertains spatial clustering of locations using high (hot spot) or low values (cold spot) with statistically significance ascertained by use of $Z$ scores and $P$ values [4–5,16]. It is calculated using the following formula:

$$G_i^* = \frac{\sum_{j=1}^n w_{ij} x_j - \bar{x} \sum_{j=1}^n w_{ij}}{s \sqrt{\frac{\left[ n \sum_{j=1}^n w_{ij}^2 - \left( \sum_{j=1}^n w_{ij} \right)^2 \right]}{n-1}}}$$

If the value $Gi^*$ is greater than 0, it indicates that the neighbor attribute value of its spatial unit $i$ is high; otherwise, the neighbor attribute value is low.

## Spatio-temporal clustering analysis

Spatio-temporal statistics were employed to describe the temporal and spatial distribution of MT-ZVL, and to identify geographic and temporal clusters, of disease within the 2015 to 2019 time period. A Poisson model using a retrospective space-time permutation scan statistic was used to identify spatio-temporal clusters of MT-ZVL using SatScan software version 9.4.2[23]. Space time scanning defined as a dynamic scan using a cylindrical window in dimensions of time scales and geographical locations, was also used in the identification of spatial and temporal disease clusters. The log likelihood ratio (LLR), a statistic which tests the difference between observed and expected numbers, in and outside the window, was employed as a measure of change in the time and space in window [24]. A Monte Carlo simulation was used for permutation testing. Statistical significance was determined by a $p$-value of $< 0.05$. Spatial clustering of MT-ZVL incidence was detected based a Poisson model using a flexible spatial scan statistic in FlexScan software, version 3.1.2 [24]. All incidence data were processed separately for each year for which data were available and potential spatial clusters detected using restricted log likelihood ratio (RLLR). $P$-values for RLLR were calculated, and the most likely cluster (MLC) estimated. A $p$-value of $< 0.05$ was indicative of a statistically significant cluster.

## Results

### Epidemic trend of MT-ZVL from 2015 to 2019

A total of 529 MT-ZVL cases were detected in Gansu, Shaanxi, Sichuan, Shanxi, Henan and Hebei provinces between 2015 and 2019, with annual incidence of 82, 95, 113, 117 and 112 cases each year, respectively (Fig 1). Among all MT-ZVL cases reported in the six provinces during the 5-year period, the highest number of cases was reported in Gansu province (55.0%), followed by Shanxi (21.7%), Shaanxi (12.5%), Sichuan (8.9%) and Henan (1.5%), with the lowest number reported in Hebei province (0.4%).

The counties (districts) with the greatest cumulative incidence of MT-ZVL were located predominantly in the Gansu-Sichuan, Shaanxi-Shanxi and Shanxi-Hebei-Henan border areas. In addition to this, 31 MT-ZVL cases were detected in 13 re-emergent counties (districts) of Gansu, Shaanxi, Shanxi, Henan and Hebei provinces, with the majority of cases in Shaanxi (9 cases) and Shanxi provinces (11 cases; Fig 2).

Joinpoint regression analysis revealed a decline in total VL incidence ($t = -5.66$, $P < 0.05$), and an increase in MT-ZVL incidence in China during the 2015 to 2019 time period ($t = 4.87$, $P < 0.05$). A significant change in MT-ZVL incidence was observed in Shanxi province during the 5-year period ($t = 16.91$, $P < 0.05$), however, no significant changes were detected in MT-ZVL incidence in other five provinces (Table 1, Figs 3–5).

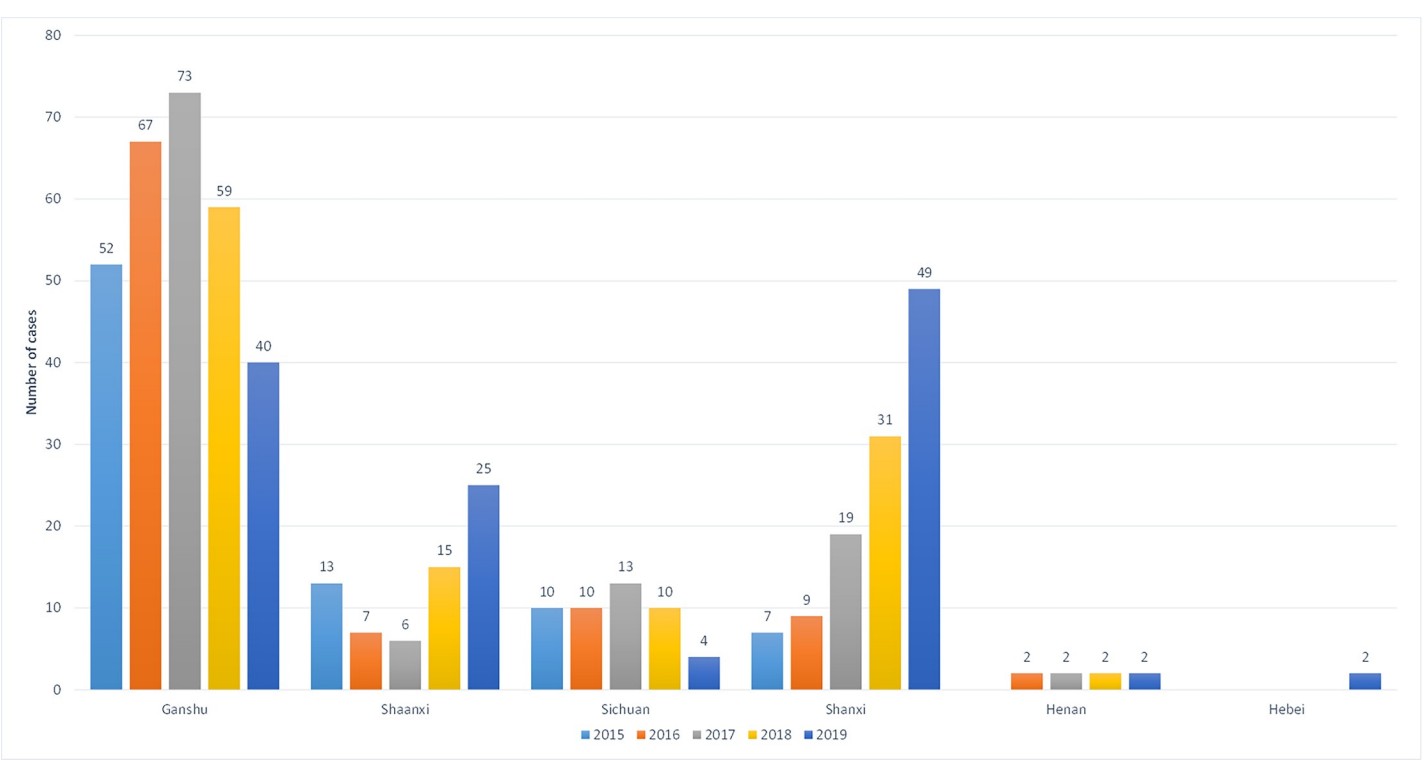

**Fig 1. The number of local infected patient of MT-ZVL in 6 provinces of Gansu, Shaanxi, Sichuan, Shanxi, Henan and Hebei, China from 2015 to 2019.**

### Global spatial autocorrelation and hotspots of MT-ZVL incidence

Positive spatial autocorrelation was observed using the Global Moran's *I* statistic among different counties (districts) in each province ($P < 0.05$; Table 2).

Hotspots of MT-ZVL infection were detected in 16, 10, 12, 11 and 29 counties (districts) in Gansu, Shaanxi, Sichuan, Shanxi, Henan and Hebei, respectively, with highest-incidence cluster defined as hotspots detected with a 99% confidence interval. MT-ZVL hotspots were identified predominantly in the southern Gansu province, northern Sichuan province and central Shaanxi province in 2015, the southern Gansu-Sichuan border areas in 2016, and in the southeastern Shaanxi province in 2017. Infection hotspots were also detected in the southern Gansu-Sichuan border areas, the eastern Shanxi province in 2018, and were widely identified in Gansu-Sichuan-Shaanxi border areas, eastern Shaanxi-Shanxi border areas and local areas of the eastern Shanxi province in 2019 (Figs 6–10).

### Spatio-temporal clusters of MT-ZVL incidence

Based on the Satscan soft analysis, retrospective space-time permutation scan statistics also detected statistically significant clusters of infection at county level in each of the six study provinces, with 3, 2, 3, 3 and 6 clusters detected for each year within the 2015 to 2019 time period, respectively, (Table 3). The degree of infection clustering was also observed to be reduced during successive years of the study period using LLR estimates. During the 2015 to 2018 study period, grade I clusters of MT-ZVL incidence were identified in the southern Gansu province, while in 2019, grade I clusters were detected only in the eastern Shanxi province. For the other clusters, they were detected in the central Sichuan province and Shaanxi-Shanxi border areas in 2015, in the central Sichuan province in 2016 and 2017. in the eastern

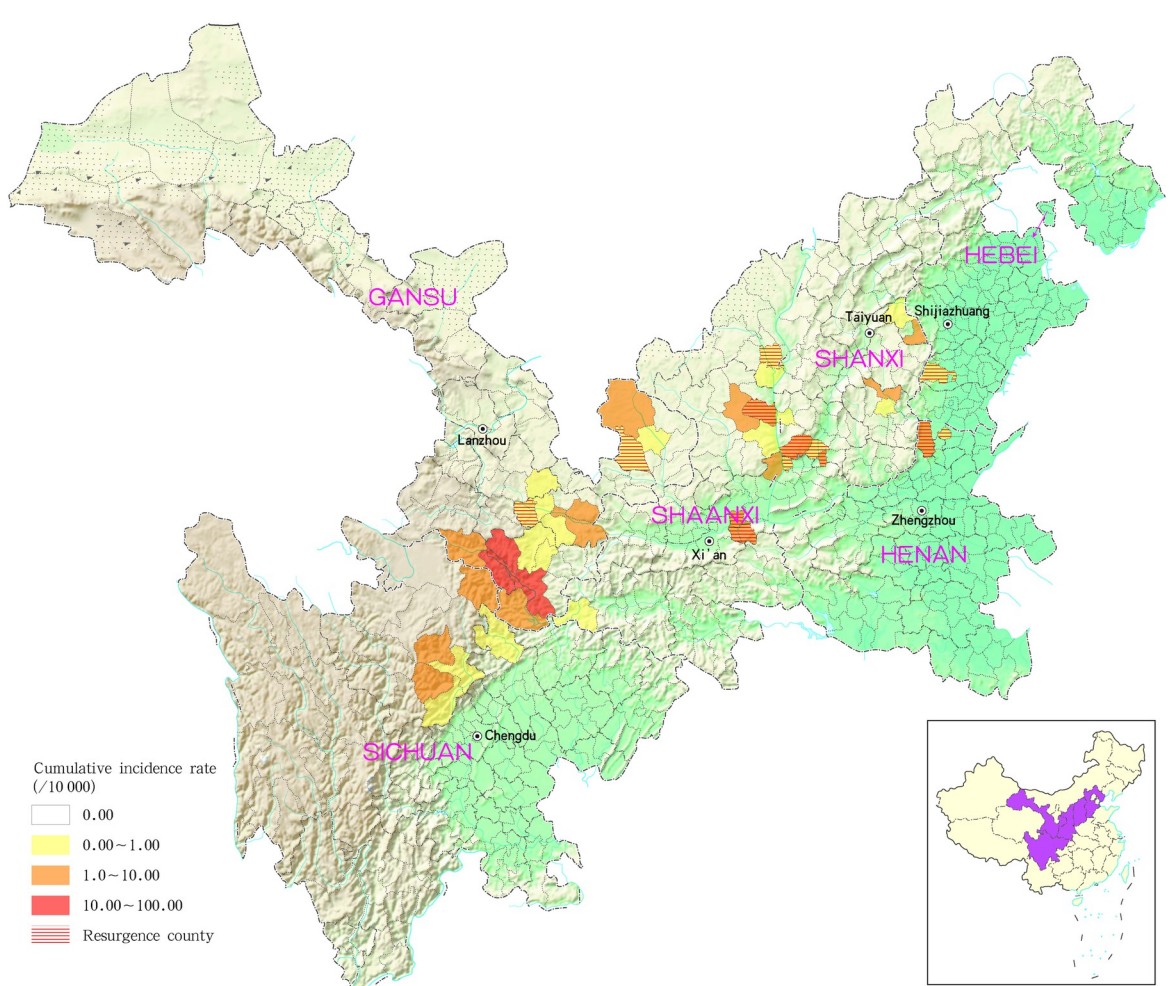

**Fig 2. Cumulative incidence and reemergence of mountain-type zoonotic visceral leishmaniasis in China from 2015 to 2019.**

parts of Shaanxi province and eastern parts of Shanxi province in 2018, and in southern Gansu province, southeastern Shaanxi province and Shaanxi-Shanxi border areas in 2019, respectively. The clustering regions and their grade were presented with different circles and colors in the Figs 11–15.

**Table 1. Joinpoint regression analysis of mountain-type zoonotic visceral leishmaniasis incidence in 6 provences, China from 2015 to 2019.**

|  | T value | *P* value | APC | Lower 95% CI | Upper 95% CI |
|---|---|---|---|---|---|
| Ganshu | -0.71 | 0.53 | -5.70 | -27.30 | 22.30 |
| Shaanxi | 1.60 | 0.21 | 23.70 | -18.90 | 88.60 |
| Sichuan | -0.84 | 0.46 | -10.10 | -39.90 | 34.40 |
| Shanxi | 16.91 | 0.00 | 65.61 | 50.60 | 82.10 |
| Henan | 0.91 | 0.43 | 13.00 | -26.30 | 73.10 |
| Hebei | 2.70 | 0.07 | 47.80 | -6.80 | 134.30 |
| six provences | 4.87 | 0.02 | 9.62 | 3.20 | 16.40 |
| China | -5.66 | 0.01 | -26.24 | -37.80 | -12.50 |

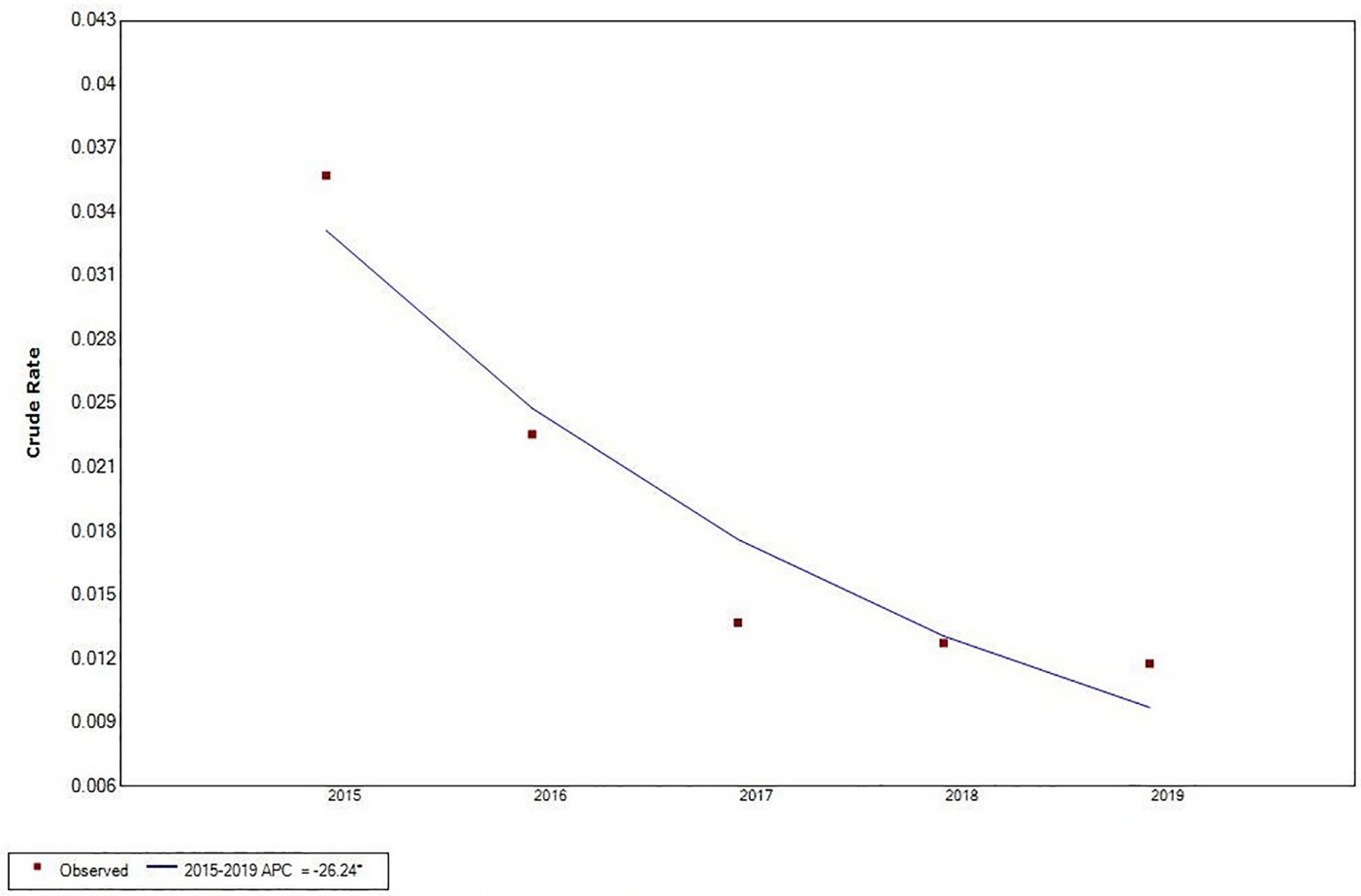

**Fig 3. The joinpoint regression analysis for determining changes in the trend of visceral leishmaniasis incidence in China from 2015 to 2019.**

For each year during the study time period, 3, 2, 4, 4 and 4 statistically significant clusters of MT-ZVL were identified in each of the six study provinces, respectively, based on flexible spatial scan statistics (Table 4). Most likely clusters were identified predominantly in the southern Gansu province during the five year study period. Secondary clusters of infection were detected in the central Sichuan province and eastern Shaanxi province in 2015, Gansu-Sichuan border areas and central Sichuan province in 2016, Gansu-Sichuan border areas, Shaanxi—Shanxi border areas, eastern parts of Shanxi province in 2017 and 2018, and in the Shaanxi-Shanxi border areas and eastern Shanxi province in 2019 (Figs 16–20).

## Discussion

As major parasitic diseases control has been reinforced by the central government of China and special funds have been given to the VL control program [25], VL transmission has been under effective control in China. Recently, however, cases of MT-ZVL have re-emerged in multiple endemic areas of China, with a gradual increase in cases each year [26]. The present study, therefore, aimed to retrospectively analyze the epidemiology, and identify the spatio-temporal distribution, of MT-ZVL in Gansu, Shaanxi, Sichuan, Shanxi, Henan and Hebei

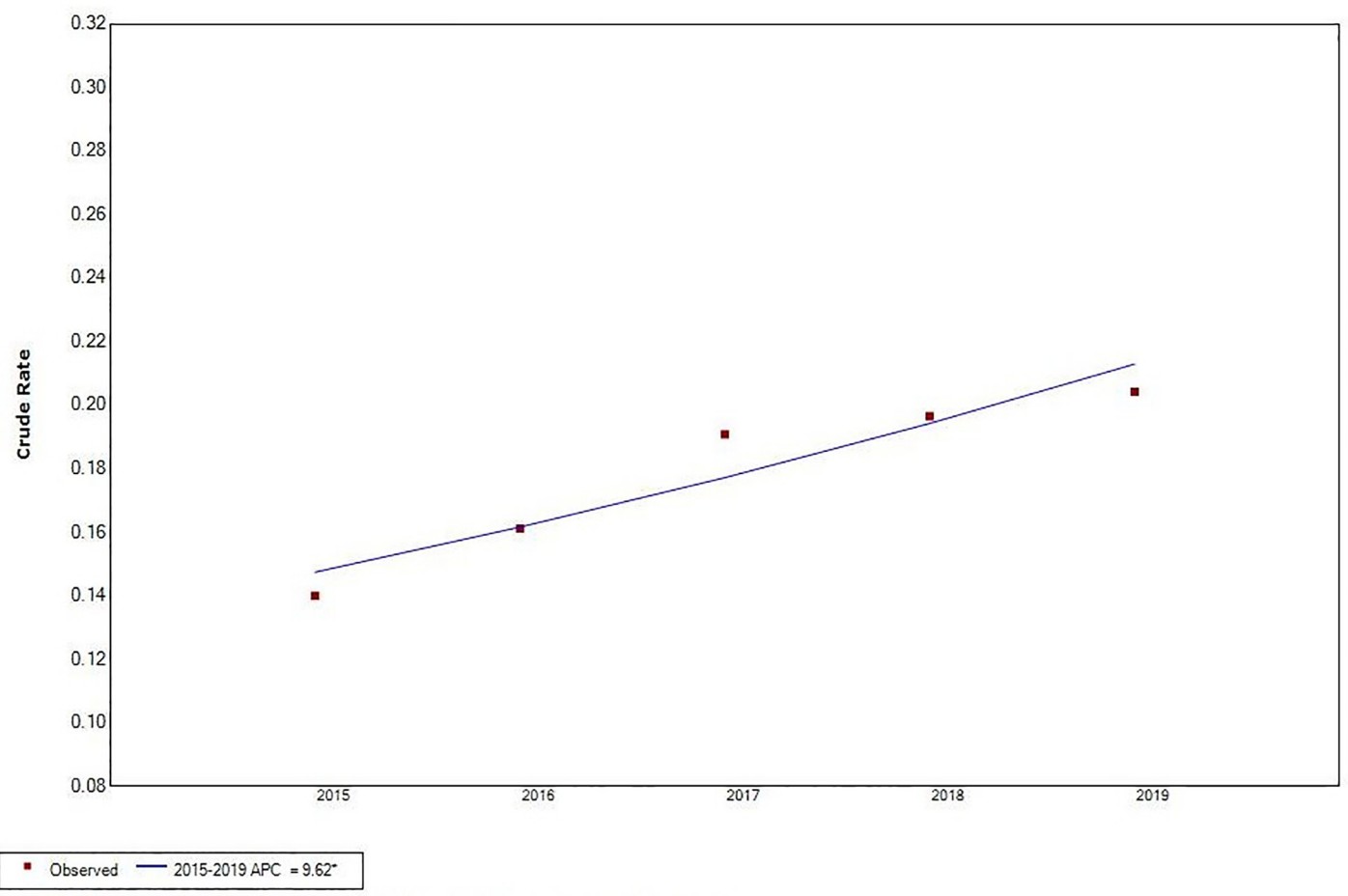

**Fig 4. The joinpoint regression analysis for determining changes in the trend of mountain-type zoonotic visceral leishmaniasis incidence in main endemic areas (6 provinces of Gansu, Shaanxi, Sichuan, Shanxi, Henan and Hebei) from 2015 to 2019.**

provinces between 2015 and 2019, in order to explore the spatio-temporal within the specified time period.

Between 2015 and 2019, a total of 529 MT-ZVL cases were reported in endemic areas of China, with the majority of cases identified in Gansu (55.0%), Shanxi (21.7%), Shaanxi (12.5%) and Sichuan (8.9%), indicating rebounded resurgence of MT ZVL in the Shanxi and Shaanxi provinces. Recently, MT-ZVL have been detected in the historically endemic and non-endemic areas of Henan and Hebei provinces, indicating re-emergence, and emergence of MT-ZVL in these two provinces. Joinpoint regression analysis determined a decline in VL incidence in China during the 2015 to 2019 study period, but an increase in MT-ZVL incidence ($t = 4.87$, $P < 0.05$), most notably significant in Shanxi province ($t = 16.91$, $P < 0.05$). These findings demonstrate that transmission of MT-ZVL is increasing, rather than declining, in some part of China, with an observable increase in MT-ZVL transmission during the past three years, and re-emergence of MT-ZVL in multiple transmission-controlled areas [26].

Spatio-temporal analysis show higher consistent hotspots and clustering regions by different methods, especially for the high risk region(grade I), using Global Moran's *I* statistic,

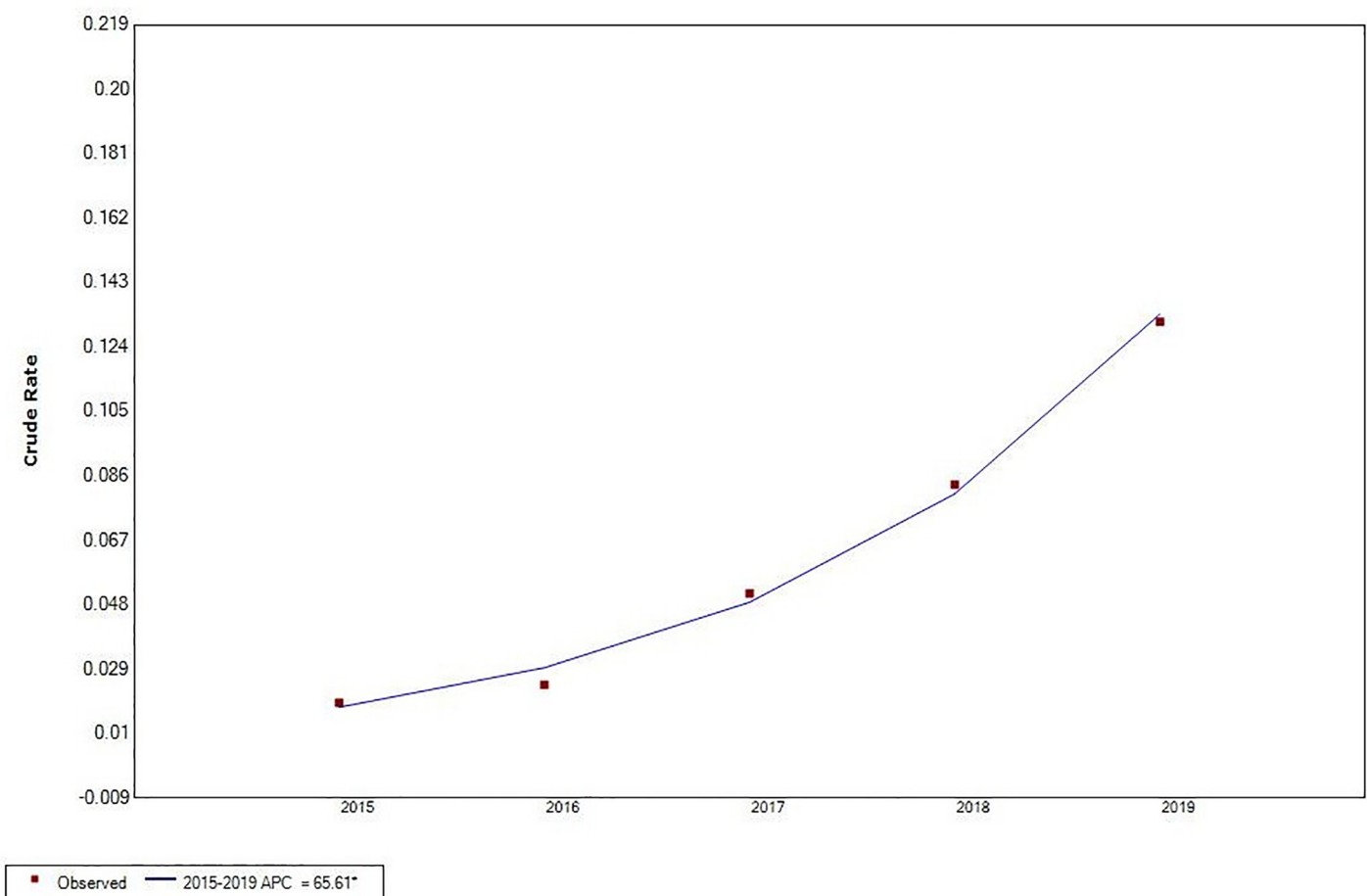

**Fig 5. The joinpoint regression analysis for determining changes in the trend of mountain-type zoonotic visceral leishmaniasis incidence in Shanxi province from 2015 to 2019.**

revealed spatial clustering of MT-ZVL in each of the six study provinces for each year of the study period ($P < 0.05$). Spatial hotspot analysis revealed infection clustering in 16, 10, 12, 11 and 29 counties (districts) in Gansu, Shaanxi, Sichuan, Shanxi, Henan and Hebei provinces respectively, with increased incidence during the past three years. Among those hotspots, they have always been found in the southern Gansu province and northern Sichuan province, and

**Table 2. Global autocorrelation analysis of mountain-type zoonotic visceral leishmaniasis incidence in 6 provences, China from 2015 to 2019.**

| Year | Moran's $I$ | Variance | Expected value | $Z$ value | $P$ value |
|------|-----------|----------|----------------|---------|---------|
| 2015 | 0.056441 | 0.000089 | -0.001196 | 6.098942 | <0.05 |
| 2016 | 0.050108 | 0.000072 | -0.001196 | 6.030147 | <0.05 |
| 2017 | 0.055217 | 0.000071 | -0.001196 | 6.675749 | <0.05 |
| 2018 | 0.048587 | 0.000054 | -0.001196 | 6.756551 | <0.05 |
| 2019 | 0.087472 | 0.000102 | -0.001196 | 8.786115 | <0.05 |

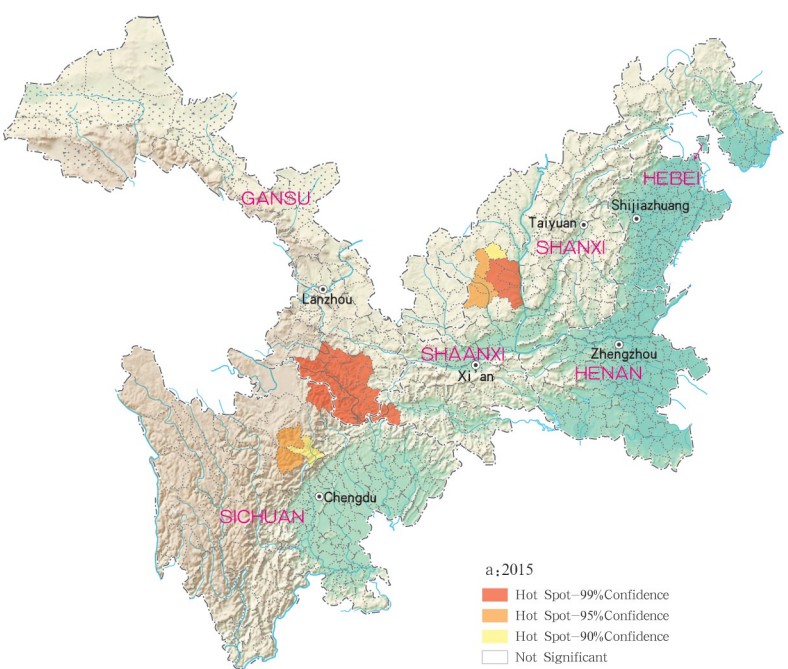

**Fig 6. Hotspot analysis of mountain-type zoonotic visceral leishmaniasis incidence in 6 provinces, China in 2015.**

multiple hotspots were detected in southern and southwestern parts of Shaanxi province, and in local regions of southwestern and eastern Shanxi province during later three years. Retrospective space-time scanning analysis identified 3, 2, 3, 3 and 6 clusters of MT-ZVL incidence in Gansu, Shaanxi, Sichuan, Shanxi, Henan and Hebei between 2015 and 2019, which

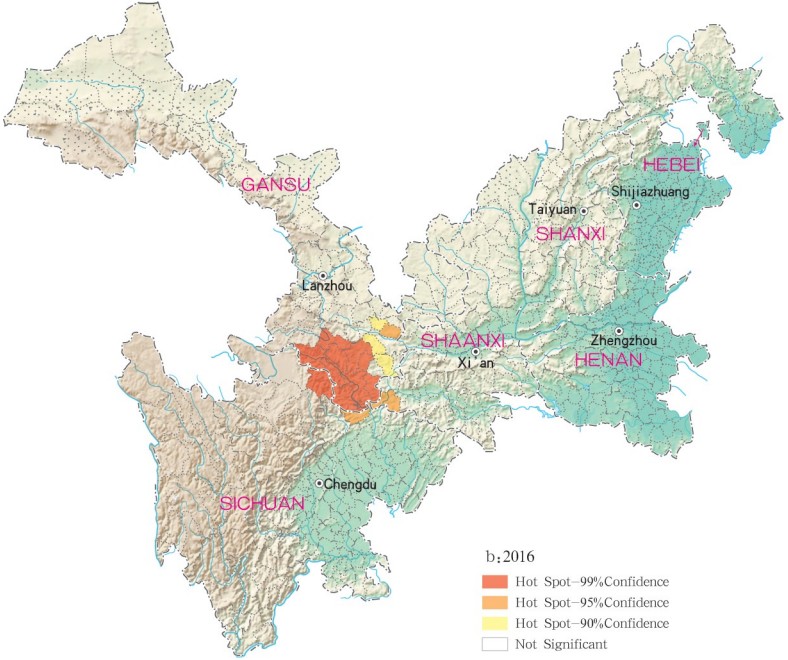

**Fig 7. Hotspot analysis of mountain-type zoonotic visceral leishmaniasis incidence in 6 provinces, China in 2016.**

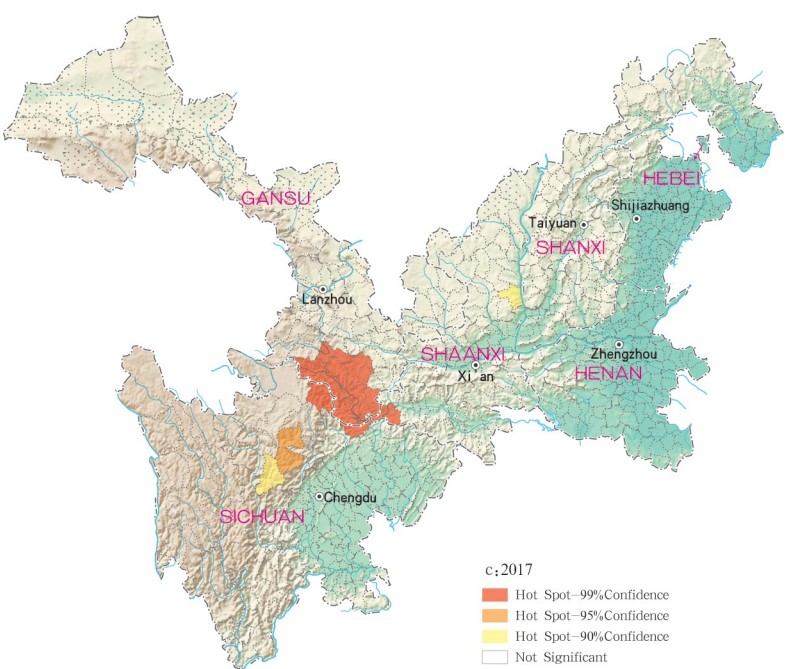

**Fig 8. Hotspot analysis of mountain-type zoonotic visceral leishmaniasis incidence in 6 provinces, China in 2017.**

corresponded to spatial-temporal distribution determined by hotspot analysis. Two grade I clusters were detected in southern Gansu province and eastern Shanxi province, demonstrating that Yangquan city, in eastern Shanxi province, is a recent high-risk region of MT-ZVL, in addition to the southern Gansu province. Flexible spatial scan statistic identified 3, 2, 4, 4 and

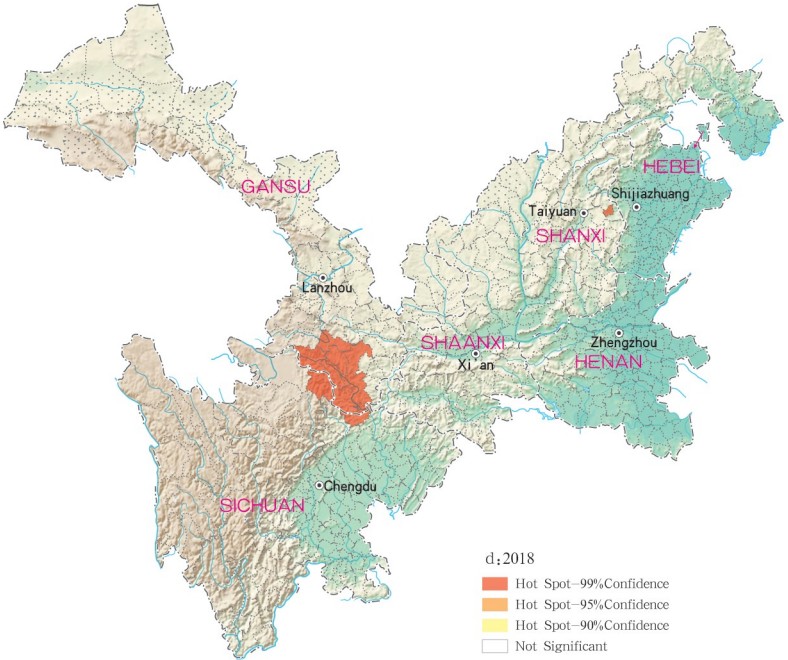

**Fig 9. Hotspot analysis of mountain-type zoonotic visceral leishmaniasis incidence in 6 provinces, China in 2018.**

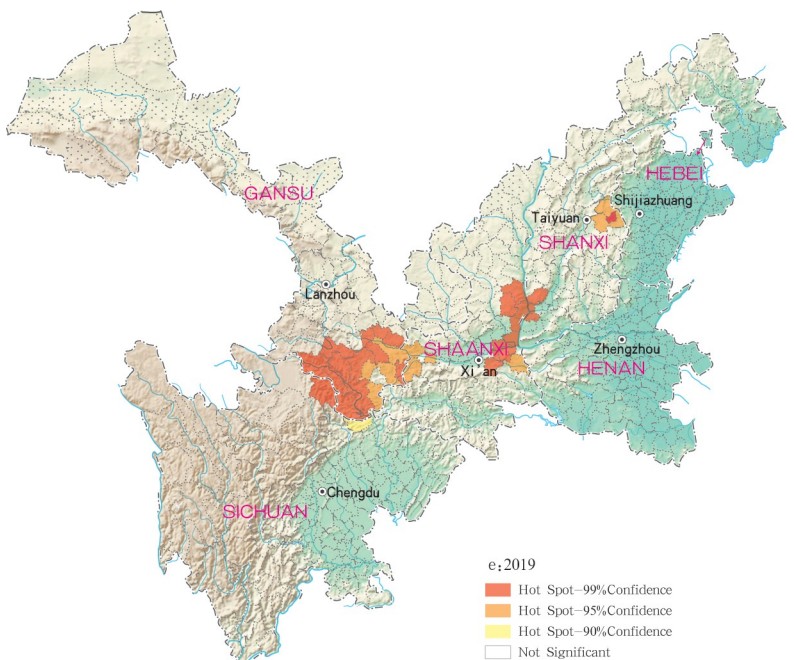

**Fig 10. Hotspot analysis of mountain-type zoonotic visceral leishmaniasis incidence in 6 provinces, China in 2019.**

4 clusters of MT-ZVL incidence in Gansu, Shaanxi, Sichuan, Shanxi, Henan and Hebei provinces between 2015 and 2019, with similar annual MLCs identified predominantly in southern Gansu province. Secondary clusters varied between years, however, with a gradual shift observed from southern Gansu and northern Sichuan province to Shaanxi-Shanxi border areas and eastern Shanxi province.

**Table 3. Spatiotemporal clustering analysis of mountain-type zoonotic visceral leishmaniasis incidence in 6 provences, China from 2015 to 2019.**

| Year | Cluster center(°) | | Radius(km) | No. of clustered counties | No. of observed | No. of expected | Relative risk | LLR | P value |
|---|---|---|---|---|---|---|---|---|---|
| | Latitude | Longitude | | | | | | | |
| 2015 | 33.6298 | 104.3176 | 83.90 | 5 | 50 | 0.56 | 228.65 | 195.01 | <0.05 |
| | 36.0647 | 110.1792 | 74.14 | 10 | 11 | 0.87 | 14.42 | 18.42 | <0.05 |
| | 31.5754 | 103.0129 | 65.63 | 4 | 8 | 0.39 | 22.08 | 16.98 | <0.05 |
| 2016 | 33.6298 | 104.3176 | 83.90 | 5 | 64 | 0.64 | 302.11 | 259.75 | <0.05 |
| | 32.1609 | 103.0473 | 65.15 | 3 | 7 | 0.33 | 22.56 | 14.88 | <0.05 |
| 2017 | 33.6298 | 104.3176 | 83.90 | 5 | 69 | 0.77 | 229.48 | 269.26 | <0.05 |
| | 37.8777 | 113.5332 | 6.06 | 3 | 10 | 0.39 | 27.79 | 23.17 | <0.05 |
| | 31.5754 | 103.0129 | 65.63 | 4 | 6 | 0.53 | 11.82 | 9.19 | <0.05 |
| 2018 | 33.6298 | 104.3176 | 83.90 | 5 | 59 | 0.79 | 148.86 | 213.87 | <0.05 |
| | 37.8777 | 113.5332 | 19.99 | 4 | 23 | 0.66 | 43.38 | 61.76 | <0.05 |
| | 35.5764 | 110.3812 | 0.00 | 1 | 8 | 0.14 | 61.54 | 24.81 | <0.05 |
| 2019 | 37.8777 | 113.5332 | 19.99 | 4 | 33 | 0.68 | 65.74 | 100.33 | <0.05 |
| | 33.6298 | 104.3176 | 83.90 | 5 | 34 | 0.83 | 56.54 | 98.16 | <0.05 |
| | 35.5764 | 110.3812 | 0.00 | 1 | 16 | 0.15 | 126.56 | 60.47 | <0.05 |
| | 34.4076 | 109.8007 | 27.55 | 3 | 9 | 0.21 | 45.38 | 25.20 | <0.05 |
| | 34.5278 | 110.0465 | 45.24 | 5 | 9 | 0.36 | 26.98 | 20.67 | <0.05 |
| | 35.9191 | 110.9317 | 40.15 | 6 | 9 | 1.00 | 9.66 | 12.07 | <0.05 |

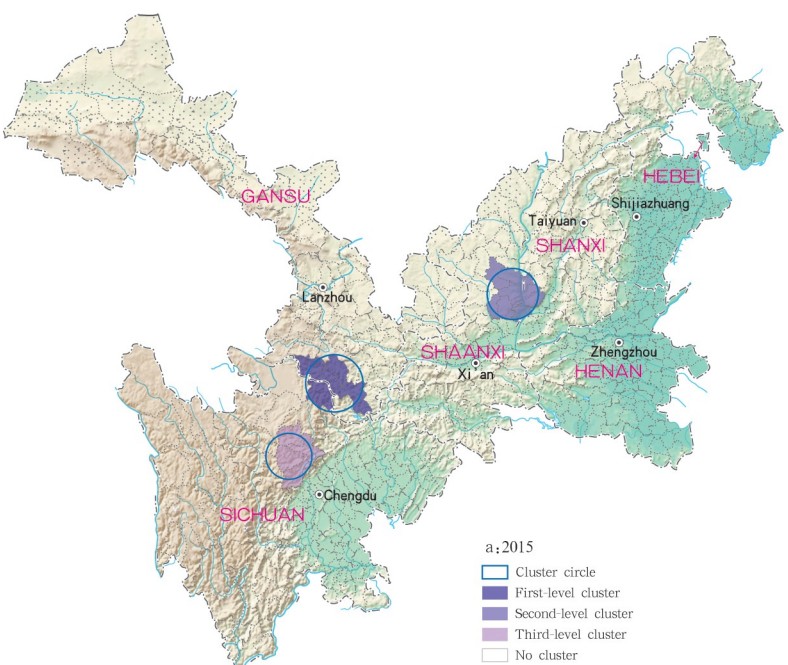

**Fig 11. Spatio-temporal clustering analysis of mountain-type zoonotic visceral leishmaniasis incidence in 6 provinces, China in 2015.**

Reinforced control efforts have resulted in a substantial decline in number of VL cases in China during the 2015 to 2019 time period [26], however, transmission of MT-ZVL appears to be increasing, rather than declining. Although sporadic outbreaks of MT-ZLV appear in

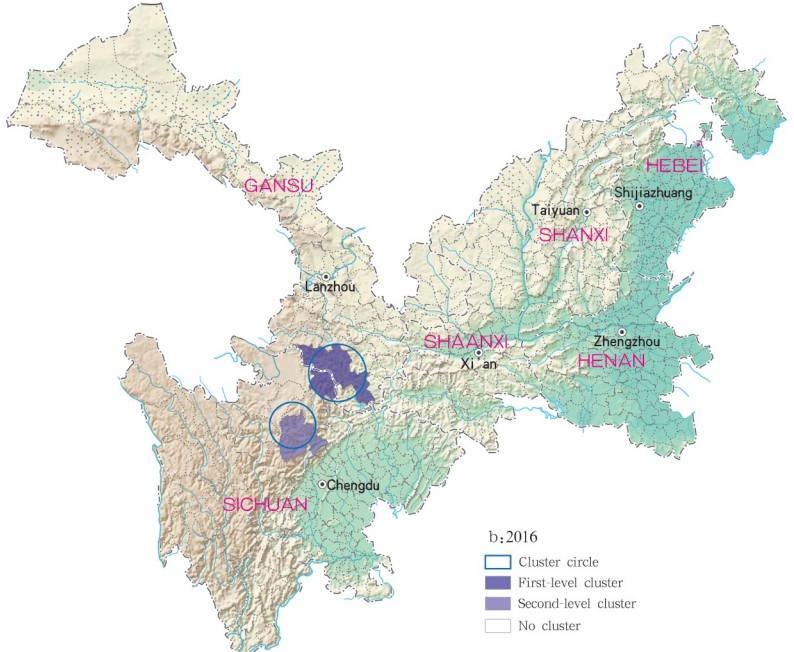

**Fig 12. Spatio-temporal clustering analysis of mountain-type zoonotic visceral leishmaniasis incidence in 6 provinces, China in 2016.**

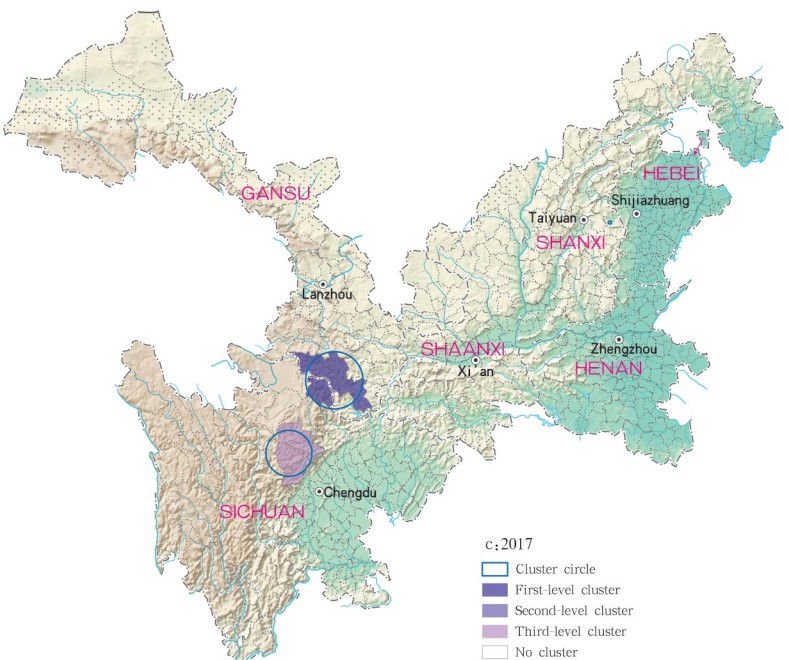

**Fig 13. Spatio-temporal clustering analysis of mountain-type zoonotic visceral leishmaniasis incidence in 6 provinces, China in 2017.**

China, there has been a substantial increase in the MT-ZVL epidemics during the past three years [11], with an increase in high-incidence clusters in localised regions [27]. Although the majority of MT-ZVL clusters were detected the southern Gansu and northern Sichuan

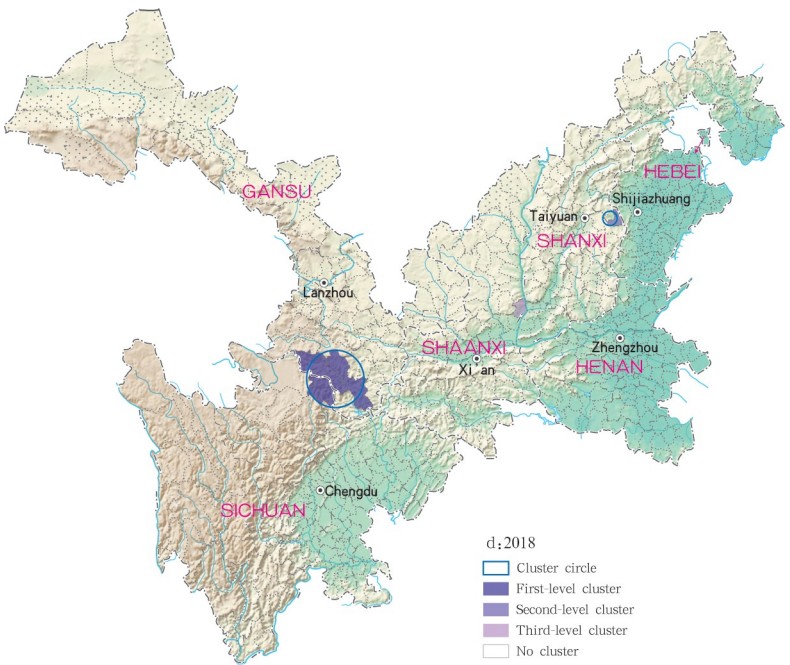

**Fig 14. Spatio-temporal clustering analysis of mountain-type zoonotic visceral leishmaniasis incidence in 6 provinces, China in 2018.**

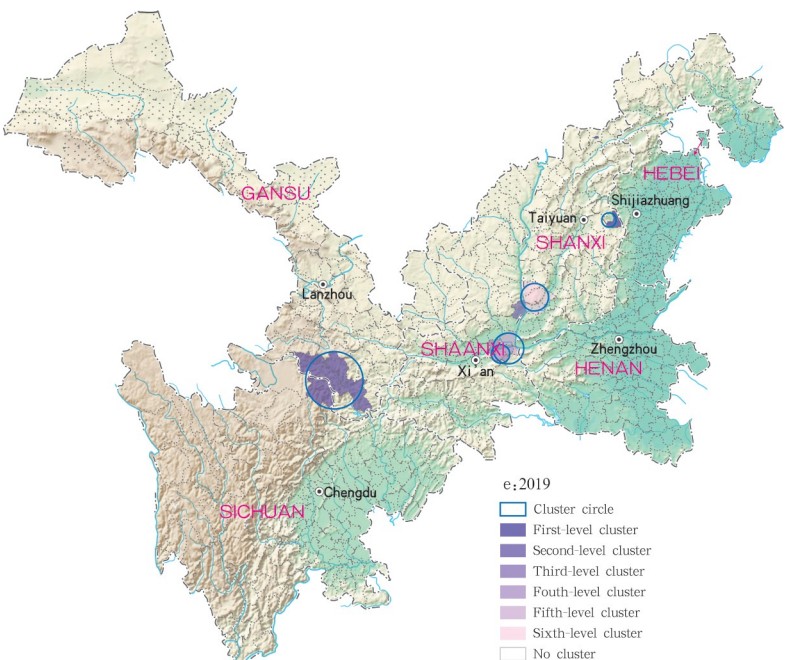

**Fig 15. Spatio-temporal clustering analysis of mountain-type zoonotic visceral leishmaniasis incidence in 6 provinces, China in 2019.**

provinces [28], regions traditionally at higher risk of MT-ZVL infection, re-emergence of MT-ZVL epidemics have also recently been detected in Shanxi, Henan and Hebei provinces, where MT-ZVL transmission had been controlled [11]. These areas are extension regions of the Loess Plateau and are mainly hilly settings, where the Yanshan-Taihangshan mountain deciduous broad-leaved forest ecological zone and Fenwei Basin Agro-ecological zone are located [29]. As temperate continental monsoon climate-covered regions, it is hot and rainy in summer, and the hilly and frondent environments provide a favorable condition for the breeding and reproduction of wild sandflies. Moreover, local loess cave dwellings and dwellings made of cement, bricks and tiles provide a suitable habitat for sandfly breeding grounds [30]. Infected animals also carry a higher risk of VL transmission and re-emergence in localised areas [31]. As a result of sustained periods of neglected VL, diagnosis, screening and management of this disease have been weakened, which also may have contributed to resurgence of VL transmission[32]. Therefore, the reinforcement of VL control in conventionally high-risk areas, attention to areas where VL re-emergence is likely, timely survey of vectors, assessment of transmission risk, and targeted interventions are strongly recommended to reduce risk of MT-ZVL infection.

**Table 4. Spatial clustering analysis of mountain-type zoonotic visceral leishmaniasis incidence in 6 provences, China from 2015 to 2019.**

| Year | No. of clusters | Most likely clusrer | | | | | |
|------|-----------------|---------------------|--|--|--|--|--|
| | | No. of counties | Max distance(km) | No. of infected cases | No. of expected cases | Overall relative risk | P value |
| 2015 | 3 | 4 | 164.95 | 48 | 0.21 | 226.07 | <0.05 |
| 2016 | 2 | 3 | 101.88 | 59 | 0.19 | 316.23 | <0.05 |
| 2017 | 4 | 4 | 164.95 | 63 | 0.29 | 297.80 | <0.05 |
| 2018 | 4 | 3 | 101.88 | 50 | 0.23 | 217.60 | <0.05 |
| 2019 | 4 | 4 | 164.95 | 33 | 0.32 | 104.46 | <0.05 |

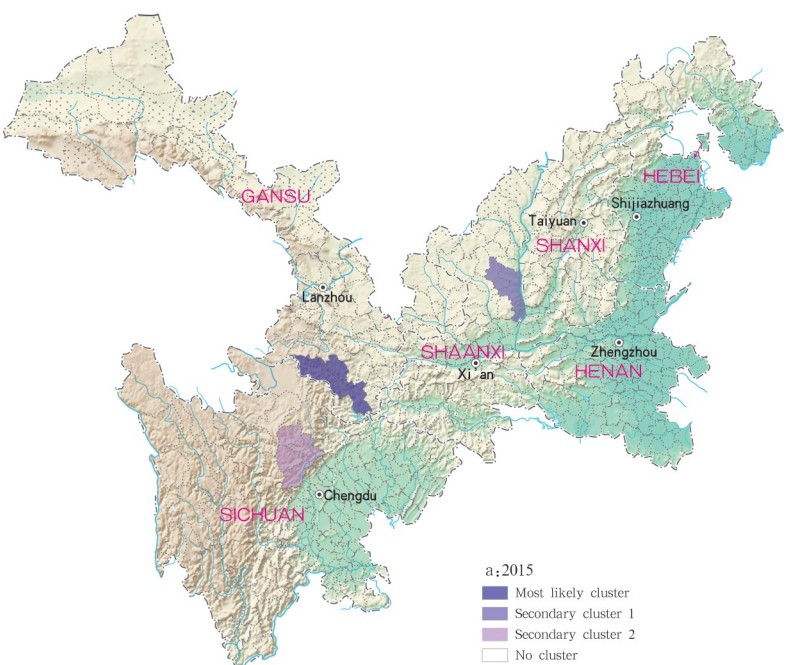

**Fig 16. Spatial clustering analysis of mountain-type zoonotic visceral leishmaniasis incidence in 6 provinces, China in 2015.**

There are also some limitations in this study. First, MT-ZVL mainly occurs in remote rural areas, and some cases may not go to the doctor in time due to mild symptoms or traffic restrictions. These cases were not reported to the local CDC, so the situation of MT-ZVL may be

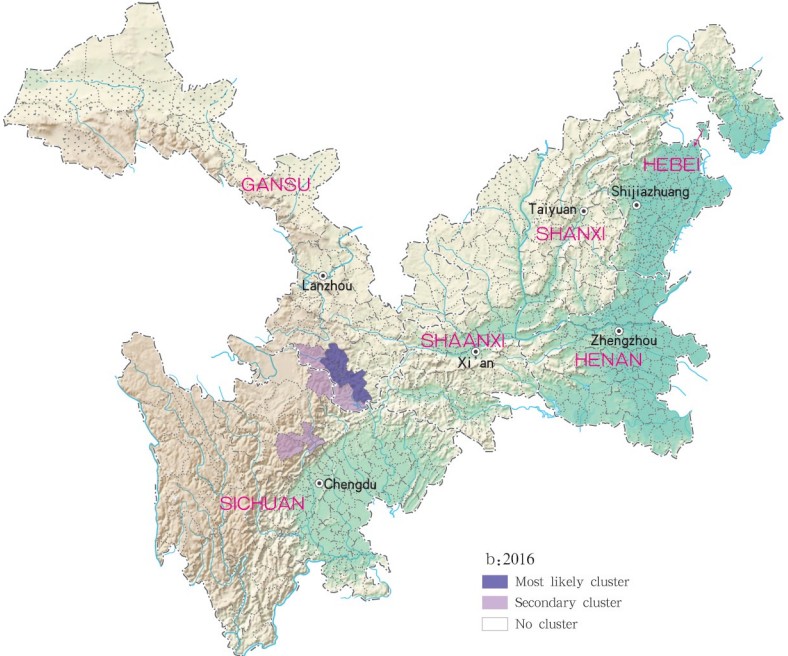

**Fig 17. Spatial clustering analysis of mountain-type zoonotic visceral leishmaniasis incidence in 6 provinces, China in 2016.**

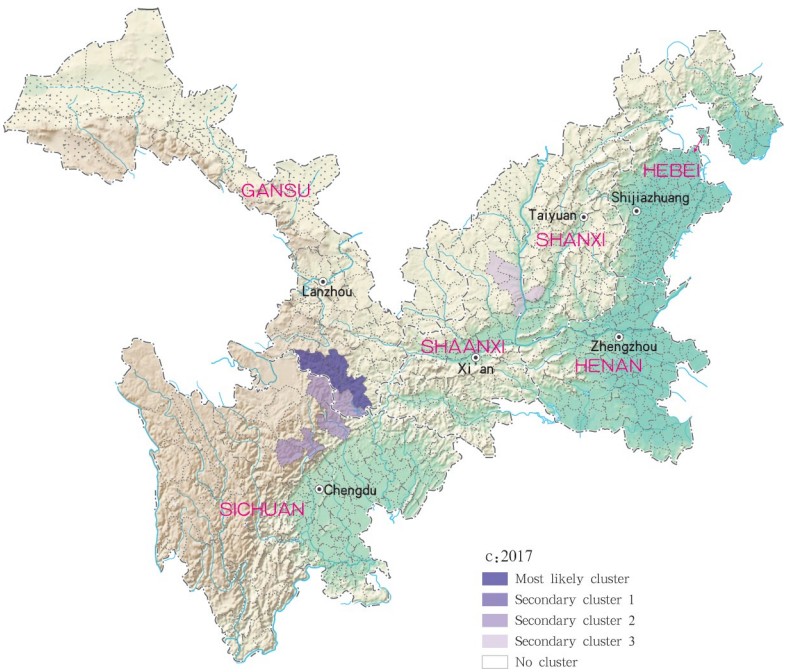

**Fig 18. Spatial clustering analysis of mountain-type zoonotic visceral leishmaniasis incidence in 6 provinces, China in 2017.**

underestimated in this study. Secondly, this study focuses on MT-ZVL re-emergence and clustering areas, and there is no definite conclusion on the causes of high cluster in local areas. In future studies, accurate case data can be obtained through field investigations, and the reasons

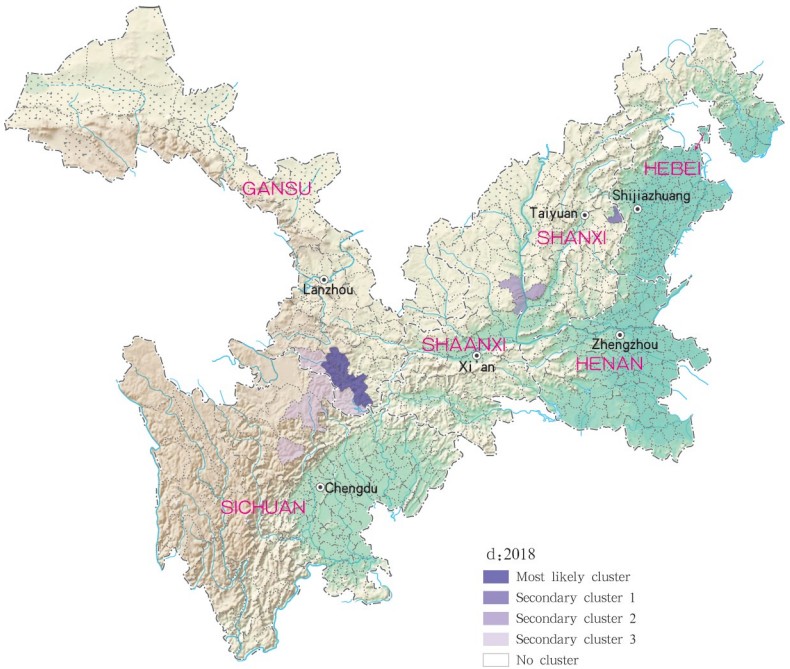

**Fig 19. Spatial clustering analysis of mountain-type zoonotic visceral leishmaniasis incidence in 6 provinces, China in 2018.**

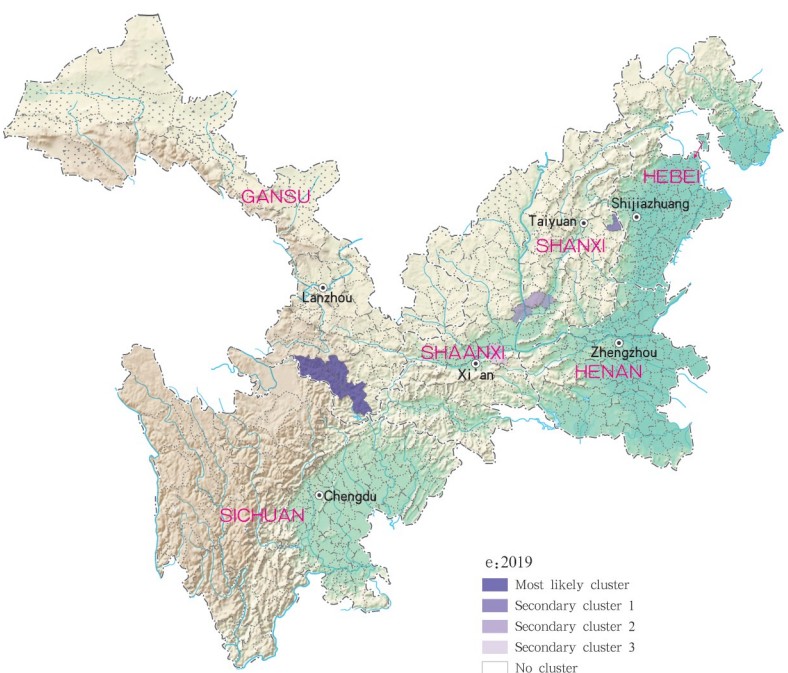

**Fig 20. Spatial clustering analysis of mountain-type zoonotic visceral leishmaniasis incidence in 6 provinces, China in 2019.**

for high cluster in local areas can be further studied, so as to provide further technical support for the VL control in this region.

## Author Contributions

**Conceptualization:** Yuwan Hao, Shizhu Li.

**Data curation:** Yuwan Hao, Xiaokang Hu, Yanfeng Gong, Zhengbin Zhou, Yuanyuan Li, Shizhu Li.

**Formal analysis:** Yuwan Hao, Shizhu Li.

**Funding acquisition:** Yuwan Hao, Jingbo Xue, Zhengbin Zhou, Yuanyuan Li, Yi Zhang, Shizhu Li.

**Investigation:** Zhengbin Zhou, Yuanyuan Li.

**Methodology:** Yuwan Hao, Xiaokang Hu, Yanfeng Gong, Jingbo Xue, Shizhu Li.

**Project administration:** Qiang Wang, Yi Zhang, Shizhu Li.

**Software:** Xiaokang Hu, Yanfeng Gong, Jingbo Xue.

**Supervision:** Shizhu Li.

**Writing – original draft:** Yuwan Hao, Shizhu Li.

**Writing – review & editing:** Qiang Wang, Yi Zhang, Shizhu Li.

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
