## [Decision Letter · Decision Letter 0]

10 Nov 2020

Dear Dr. Li,

Thank you very much for submitting your manuscript "Spatio-temporal clustering of Mountain-type Zoonotic Visceral Leishmaniasis in China between 2015 and 2019" for consideration at PLOS Neglected Tropical Diseases. As with all papers reviewed by the journal, your manuscript was reviewed by members of the editorial board and by several independent reviewers. The reviewers appreciated the attention to an important topic. Based on the reviews, we are likely to accept this manuscript for publication, providing that you modify the manuscript according to the review recommendations. 

Sincerely,

Johan Van Weyenbergh

Associate Editor

Nadira Karunaweera

Deputy Editor

Reviewer's Responses to Questions

**Key Review Criteria Required for Acceptance?**

**Methods**

-Are the objectives of the study clearly articulated with a clear testable hypothesis stated?

-Is the study design appropriate to address the stated objectives?

-Is the population clearly described and appropriate for the hypothesis being tested?

-Is the sample size sufficient to ensure adequate power to address the hypothesis being tested?

-Were correct statistical analysis used to support conclusions?

-Are there concerns about ethical or regulatory requirements being met?

Reviewer #1: The statistical and models are well written but some conceptual information needs to be reviewed. For instance, it is not clear which variables were selected as input for the clustering model and why. Also, a general description of the study area is missing. Understanding the region and main attributes of the size provinces are essential for the discussion. Moreover, the mentioned data source for the samples is not included as a reference.

Reviewer #2: (No Response)

Reviewer #3: The objectives for this study are clearly articulated, and I feel that the overall study design is appropriate for the stated objectives.

The population is clearly described and appropriate for this study.

The sample size is sufficient to support the conclusions drawn.

One of the questions that I have regarding the statistical analysis used is that three different types of cluster analysis were used to describe the clustering of this disease. What is the benefit of using the different types of cluster analysis? This research did not discuss how the different methods could contrast or support each other. Also some of the terms used is the results, specifically a grade 1 cluster, were not explained before hand.

**Results**

-Does the analysis presented match the analysis plan?

-Are the results clearly and completely presented?

-Are the figures (Tables, Images) of sufficient quality for clarity?

Reviewer #1: In general, the results are disorganized. The annual incidences presented in the text do not match the total number (529) nor the labels in the graph in the first figure. It is noticeable, that whenever results are presented for the six provinces respectively, it creates confusion for the reader due to miswriting. For instance, 5 different values are written as hotspots of infection respectively for the 6 provinces. Regarding results discussion, it gets confused with the introduction. The first three paragraphs are not analyzing the results but reporting general and conceptual facts as in the introduction. The discussion needs to be rewritten. In addition, it is important to represent cities mentioned in the text on the map, therefore readers who do not know the region would not get lost in the discussion.

Reviewer #2: (No Response)

Reviewer #3: The analysis presented does match the analysis described in the methods.

The results are clearly presented, however, as mentioned in my comments for the methods, it was not made clear my three different cluster analysis methods were used when the results indicate very similar outcomes. More information either in this section or the discussion about the comparison and contrasts between the methods would be beneficial, or reducing the number of cluster analysis methods used.

Overall, the figures present the information presented, however there are some small additions that I feel would help deliver the information better. First, I would include a map of study area within the context of China as a whole for audience members that may not be familiar with the locations of Chinese provinces. Second, since the disease is a "Mountain-type" disease and the paper mentions that there are geographic and ecological variations in may be useful to include information on the elevation in the selected study area to help people unfamiliar with the provinces understand if the incidence areas are close to mountains or not. Third, some minor clarifications and improvements to the maps presented would present a clearer picture of the results. I think that the figures in general should include a neat line around each individual map, that the word "legend" should be removed from the legends, and the scale bars should be standardized to either 500 km or 1000 km. Fourth, in Figure 5, a-e, it is not readily apparent what the circles on the map are indicating. Is this a part of the SatScan analysis used? In the main body of the text there is no indication of what these circles mean.

**Conclusions**

-Are the conclusions supported by the data presented?

-Are the limitations of analysis clearly described?

-Do the authors discuss how these data can be helpful to advance our understanding of the topic under study?

-Is public health relevance addressed?

Reviewer #1: The conclusion does not make a clear statement of limitations, recommendations, or relevance of the study. Neither supports the article's objective. It is included in the objective the will to indicate important insights identified through the study for developing interventions aimed at the disease in question. However, nothing is mention in the conclusion.

Reviewer #2: (No Response)

Reviewer #3: There is material in the conclusion section in the general discussion of MT-ZVL that I feel would be better supported in the introduction of the paper rather than in the conclusions. Specifically, I feel that the second to fifth paragraphs, starting with "Currently, there are three types of VL in China..." and continuing for the next two paragraphs, feels like it should be included in the introduction as a general description of the disease. Aside from this, I feel that the conclusions are supported by the data and analysis presented, although a discussion of what differences are coming from the different cluster analyses used would be helpful. 

The limitations are clearly described.

The authors do discuss the relevance of the analysis to the current medical field and addresses the relevance to public health.

**Editorial and Data Presentation Modifications?**

Reviewer #1: The map panels must be reviewed. It is not clear, neither on the image nor in the figure's description, the referenced year for each map. For maps with the same legend (eg. Fig 4), I suggest adding a bigger legend only once. Colors' differences between groups in Figures 5 and 6 are not clear - different colors would be better for highlighting groups than a gradient of a unique color. It is important to standardize the clusters' names in text and figures. It is sometimes referred to as level or sometimes as a grid. Moreover, it is necessary to review the references. In general, it does not follow the right pattern for referencing digital media and websites.

Reviewer #2: (No Response)

Reviewer #3: As discussed in the results section, there are some general points about the maps that will help with the data presentation:

Overall, the figures present the information presented, however there are some small additions that I feel would help deliver the information better. First, I would include a map of study area within the context of China as a whole for audience members that may not be familiar with the locations of Chinese provinces. Second, since the disease is a "Mountain-type" disease and the paper mentions that there are geographic and ecological variations in may be useful to include information on the elevation in the selected study area to help people unfamiliar with the provinces understand if the incidence areas are close to mountains or not. Third, some minor clarifications and improvements to the maps presented would present a clearer picture of the results. I think that the figures in general should include a neat line around each individual map, that the word "legend" should be removed from the legends, and the scale bars should be standardized to either 500 km or 1000 km. Fourth, in Figure 5, a-e, it is not readily apparent what the circles on the map are indicating. Is this a part of the SatScan analysis used? In the main body of the text there is no indication of what these circles mean.

**Summary and General Comments**

Reviewer #1: The article lacks a discussion on how its results impact society and local institutions of disease control.

Reviewer #2: This is interesting research to explore the dynamics of Mountain-type Zoonotic Visceral Leishmaniasis in China between 2015 and 2019. The authors did a great work to analyze the data by using some modern analysis methods. The manuscript is well written and acceptable. However, I have some comments for the authors.

1.In material and methods section, how long was the time interval in your data acquisition? Monthly or Yearly? If the MT-ZVL incidence is annual data, how to ensure validity in the Joinpoint model by limited values? 

T-test is not suitable because the data within a certain period of time was dependent, unless chi square test checked its independence. Or use chi square test to check incidence of the same year between 6 provinces.

2.In spatial autocorrelation analysis, which disease indicator was employed to calculate Moran’I index? It should be clarified by illustrative statements.

3.In results section, for hotspots, it should be explained more clearly in methods section about the map. And the year should be annotated in the legend.

Reviewer #3: I have no additional comments
---

## [Decision Letter · Decision Letter 1]

15 Jan 2021

Dear Dr. Li,

We are pleased to inform you that your manuscript 'Spatio-temporal clustering of Mountain-type Zoonotic Visceral Leishmaniasis in China between 2015 and 2019' has been provisionally accepted for publication in PLOS Neglected Tropical Diseases.

Best regards,

Johan Van Weyenbergh

Associate Editor

Nadira Karunaweera

Deputy Editor

---

## [Editor Report · Acceptance letter]

13 Mar 2021

Dear Dr. Li,

We are delighted to inform you that your manuscript, "Spatio-temporal clustering of Mountain-type Zoonotic Visceral Leishmaniasis in China between 2015 and 2019," has been formally accepted for publication in PLOS Neglected Tropical Diseases.

Best regards,

Shaden Kamhawi

co-Editor-in-Chief

Paul Brindley

co-Editor-in-Chief
